# *Pseudomonas* spp.: Are Food Grade Organic Acids Efficient against These Spoilage Microorganisms in Fresh Cheeses?

**DOI:** 10.3390/foods10040891

**Published:** 2021-04-19

**Authors:** Erica Tirloni, Cristian Bernardi, Simone Stella

**Affiliations:** Department of Health, Animal Science and Food Safety, University of Milan, Via dell’ Università 6, 26900 Lodi, Italy; cristian.bernardi@unimi.it (C.B.); simone.stella@unimi.it (S.S.)

**Keywords:** *Pseudomonas fluorescens*, lactic acid, acetic acid, Primo sale cheese, MIC

## Abstract

Psychrotolerant *Pseudomonas* spp. are among the most common spoilage agents in fresh, soft and semi-soft cheeses; therefore, hurdles inhibiting their growth are in strong demand by producers. This study aimed to establish Minimal Inhibiting Concentrations (MICs) of lactic and acetic acid towards *P. fluorescens* and to evaluate the efficacy of a cheese surface treatment with these two organic acids. MICs were determined in Brain Heart Infusion broth at 30 °C: the inhibition was achieved at a concentration of 49.96 mM and 44.40 mM of acetic and lactic acid, respectively. Two series of inhibition tests were performed on fresh “Primo sale” cheese, inoculated with *P. brenneri* MGM3, then dipped into different acid solutions (acetic acid: 49.96, 99.92 and 149.88 mM; lactic acid: 44.40, 88.80 and 133.20 mM) and stored at 6 °C. *P. brenneri* MGM3 were enumerated, including a control series. A significantly lower growth was revealed at the highest concentrations tested, both for acetic (*p* < 0.01) and lactic acid (*p* < 0.05) if compared to control samples. A conditioning of “Primo sale” surface with organic acid solutions could be a useful hurdle for *Pseudomonas* inhibition and shelf-life extension; it should be applied in combination with other mild interventions to fight spoilage and maintain the original product characteristics.

## 1. Introduction

A large proportion of food gets spoiled before reaching the end consumer. According to the Food and Agriculture Organization of the United Nations, one-third of the food produced for human consumption is spoiled or wasted [1]. Spoilage of food for human consumption is a world-wide issue that is of particular interest for stakeholders, food producers and consumers. Food spoilage could be defined as a loss of quality in terms of colour, odour, texture and in general a loss in sensorial characteristics [2], and could be attributed to a microbiological, chemical or physical source [3]. Food spoilage is mainly caused by microorganisms that can rapidly colonize and replicate on fresh food. Psychrotolerant *Pseudomonas* spp. are among the most common bacteria implicated in spoilage, especially of refrigerated food with a prolonged shelf life, where they are likely selected [4,5,6]. *Pseudomonas* spp. are aerobic, Gram-negative, non-spore-forming bacteria; some strains produce pigments, i.e., the yellow-green fluorescing pigment called pyoverdine (especially some *P. fluorescens* and *P. aeruginosa* strains) and the blue-green pigment called pyocyanin [7,8,9]. *Pseudomonas* spp. produce thermotolerant lipolytic and proteolytic enzymes that reduce the quality and the shelf-life of raw and processed milk [10]. These bacteria are inactivated by the thermal processes currently applied through the production of most dairy products, but they can also enter the food production chain as post-process contaminants, due to the contact of the final product with soil, water or raw material [11]. Pseudomonads are also recognized to be able to colonize environmental production as well as equipment and facilities for long periods, thanks to their ability to produce persistent biofilms [12,13]. 

Dairy products, and especially fresh cheeses made by enzymatic coagulation, very often get spoiled by microorganisms, owing to their high water activity and to the neutral pH, with *P. fluorescens* group recognized as a frequent cause of these alterations [14]. The reduction of these spoilage organisms in pasteurized dairy products is critical for extending their shelf life [15]. In this context, the use of weak organic acids may be convenient. The possibility for the undissociated form of these compounds to pass the bacterial cell membrane causes an alteration of the internal balance of bacterial cells: the decreased pH within the cytoplasm, due to the dissociation of the organic acid, requires an energy expense of the cell to pump out hydrogen ions, trying to re-establish the internal pH [16]. The addition of organic acids is allowed by the European Union legislation [17], and it may be suitable for many Ready to eat (RTE) foods [18], including fresh cheese, with the aim to reformulate the products leading to a substrate less permissive for bacterial replication.

The first aim of the present study was to establish Minimal Inhibiting Concentrations (MICs) of lactic and acetic acid towards *P. brenneri* MGM3, a member of *P. fluorescens* subgroup [19,20], isolated from a fresh cheese matrix and cultured in broth. Secondly, challenge tests with *P. brenneri* inoculation onto the surface of Primo sale cheese, an Italian fresh cheese made from pasteurized milk, were conducted: the ability of cheese dipping treatments with lactic or acetic acid in limiting the bacterial growth was evaluated. 

## 2. Materials and Methods

### 2.1. MICs Determination 

A strain of *P. brenneri* MGM3, previously isolated from a fresh dairy product (Mozzarella cheese) and identified by MALDI-TOF (Matrix-Assisted Laser Desorption-Ionisation-Time of Flight Mass Spectrometry) at the microbiology laboratory of the Department of Veterinary Medicine, University of Milan (Addis, personal communication) was used for this trial. The strains stock was kept frozen at −80 °C in Microbank Cryogenic vials (Pro-Lab Diagnostics U.K., Merseyside, UK). From the stock culture, a loopful was transferred to Brain Heart Infusion broth (BHI) (Oxoid, Basingstoke, UK) with pH 7.2, and incubated at 30 °C for 48 h. The culture was harvested in late exponential growth phase, defined as a relative change in Optical Density (OD) of 0.05–0.2 at 540 nm at 540 nm determined with a spectrophotometer (Jenway 6105, Staffordshire, UK). Cell concentrations of this pre-culture was determined by microscopy at 1000× magnification (Motic, B310, Wetzlar, Germany), considering that one cell per field of view corresponded to a concentration around 10^6^ CFU/mL [21]. 

Ten millilitres tubes containing Brain Hear Infusion (BHI) broth added with aliquots of acetic acid (code 1005706, Sigma Aldrich, Steinheim, Germany) or lactic acid (code 252476, Sigma Aldrich) were prepared for the inoculation. The following final concentrations were obtained: 2.50 mM, 5.00 mM, 12.49 mM, 24.98 mM and 49.96 mM for acetic acid, and 2.78 mM, 5.55 mM, 11.10 mM, 22.20, mM and 44.40 mM for lactic acid; a control series (CTRL: BHI broth) was also prepared. After adjustment of bacterial concentration, aliquots of 0.1 mL of *P. brenneri* suspension were inoculated into the broths, achieving a final concentration of 100 CFU/mL. At the time of inoculation (t_0_), OD was measured, and the tubes were then incubated at 30 °C in duplicate. At fixed times (24, 48, 72, 96 h from inoculation), OD was newly measured. Blank (non-inoculated) BHI broth samples were prepared for each series and incubated in the same conditions; these samples were used for the comparison of the respective inoculated samples. At t_0_, the pH of the broths was also measured.

### 2.2. Challenge Tests

Specific challenge tests with Primo sale cheese, a fresh cheese made from cows’ milk produced and immediately marketed, were carried out to evaluate the effect of different concentrations of acetic and lactic acid on *P. brenneri* MGM3 inoculated on a real cheese matrix. The concentrations tested were set taking into account the MICs determined in Section 2.1. 

Primo sale cheese samples (200 g) were taken on the first day after production. Before inoculation, the cheese was sliced, obtaining standard 8 g—weight pieces.

*P. brenneri* MGM3 stock was kept frozen at −80 °C in Microbank Cryogenic vials; from the culture, a loop was transferred to BHI and incubated at 30 °C for 48 h. In order to pre-adapt the cells to the environmental conditions of the challenge tests, the cultures were subsequently re-inoculated in BHI broth and then incubated at 6 °C. The culture was harvested as described in a previous section. Finally, the culture was diluted in sterile saline water (0.85% NaCl) and spread onto the surface of the cheese to obtain the starting concentration around 2 Log CFU/g. To minimize changes in product characteristics, the inoculum volume did not exceed 1% of the weight of the samples. 

*P. brenneri*-inoculated samples were then divided into five series, four of which were dipped in the different acid solutions (acetic acid: Ac1 = 49.96 mM, Ac2 = 99.92 mM; lactic acid: Lac1 = 44.40 mM, Lac2 = 88.81 mM); the samples of the fifth series (control, CTRL) were dipped in sterile water. All the inoculated samples were then incubated at 6 °C; during storage, temperature was recorded by data loggers (Escort iLog, Escort Data Logging System Ltd., Aesch Bei Birmensdorf, Switzerland) and analysed in triplicate for *Pseudomonas* spp. enumeration and pH determination. The sampling times were set at t_0_ and after 3 h (aiming to verify the eventual microbial inactivation due to the contact with the acid solution), 48 and 96 h from inoculation. At each sampling time, the whole cheese aliquot (8 g) was 10-fold diluted in pre-chilled sterile saline and homogenized for 60 s in a peristaltic blender BagMixer^®^ 400 W (Interscience, Saint Nom, France). Further appropriate 10-fold dilutions of the homogenates were made with pre-chilled (maintained at refrigeration temperatures) sterile saline. *P. brenneri* was enumerated by spread plating on *Pseudomonas* agar base added Cephalothin, Fucidin, Cetrimide (CFC) Supplement (Scharlab, Barcelona, Spain) and incubated at 30 °C for 48 h. An increase of +0.5 Log CFU/g was used to discriminate growth and no growth in the product. The surface pH of each cheese sample was measured with a pH meter equipped with a penetration electrode in triplicate (Amel, Milan, Italy), in order to evaluate the environmental condition of the specific niche where the microorganisms were inoculated (Amel, Milan, Italy). 

Based on the results obtained from the first trial, a second challenge test was performed on Primo sale cheese samples; the same protocol described above was applied, with some modifications. Higher organic acid concentrations were tested, namely Ac3 (acetic acid, 149.87 Mm) and Lac3 (lactic acid, 133.2 mM); as for the previous challenge test, a control series (CTRL) was included. All the inoculated samples were then incubated at 6 °C and analysed in triplicate at t_0_ and after 3, 24 and 48 h from inoculation.

### 2.3. Statistical Analysis

Data obtained from *Pseudomonas* counts (expressed as Log CFU/g) were submitted to a two-way univariate Analysis of Variance (ANOVA) in SAS (version 9.1, 2016; SAS Institute Inc., Cary, NC, USA) to reveal potential differences in the treatments. Threshold values for statistical significance were set at *p* < 0.05 and *p* < 0.01.

## 3. Results and Discussion

In dairy industry, psychrotrophic microorganisms are predominant: among all, *Pseudomonas* spp. can represent till 75% of the total bacterial count of milk [5,22]. These unwanted microorganisms are responsible for spoilage of dairy products (in fact they dominate the microflora at the end of the shelf-life), acting a crucial role with negative consequences on the quality and shelf life of processed milk [5,10,23]. *Pseudomonas* are reduced in milk by heat treatments and hygiene plans, but post-pasteurization contamination is frequent and difficult to avoid in a dairy processing plant. Fresh cheeses, like Mozzarella or Primo sale cheese, are frequently involved in *Pseudomonas* spp. contamination [9,11,24,25]. Bioprotective cultures, EDTA-Na_2_ and lysozyme have been previously studied to evaluate their antagonistic activity against *Pseudomonas* spp. [6,26], but very limited literature is available on the effect of organic acids especially when applied to fresh cheese. Organic acids, according to Reg. EU 1333 (2008) [17] are allowed for addition to several food categories with the *quantum satis* principle; thus, the estimation of MIC is particularly important to set the minimum effective concentration to be applied.

In the present study, MICs for lactic and acetic acid against *P. brenneri* MGM3 growth were calculated in order to identify the growth boundary (the concentration separating growth/no-growth conditions for organic acid concentrations). The results are reported in Table 1. Our experiments highlighted that, at 30 °C, BHI with a concentration of acetic acid of 49.96 mM did not support *P. brenneri* MGM3 growth within 72 h; the same effect was achieved with a concentration of lactic acid of 44.40mM. In recent years, some studies reported the MIC of lactic and acetic acid for pseudomonads, showing variable results. The MIC value obtained in our study for acetic acid was higher than those reported by previous studies: Cruz-Romero et al. [27] investigated the antimicrobial activity of chitosan, organic acids and nano-sized solubilisates for potential use in smart antimicrobial-active packaging for food applications, revealing a MIC of 16.7 mM for acetic acid. These data were in agreement with the study by Moreira et al. [28] who found a MIC of 0.78 µL/mL (equal to 13.63 mM) towards *P. aeruginosa*; a lower value was observed by Nakai and Siebert [29], with a MIC of 0.346 g/L (5.8 mM). Bushell et al. [30] tested *P. aeruginosa* at 0, 5, 10, 20 mM of acetic acid at different pH, finding the MIC at the concentration of 20 mM at pH 5. Also, considering lactic acid, very different results were obtained by various authors: Moreira et al. [28] detected a higher MIC value (6.25 µL/mL, equal to 83.88 mM), whereas lower values were obtained by Nakai and Siebert (0.630 g/L, 7 mM) and by Burns et al. (1.41 mg/mL, 15.7 mM) [29,31]. Different outputs were found in literature, probably due to differences related to the microorganisms involved, to the use of different substrates involved in the determination of the MIC values (broths at different pH) and to the temperatures applied in the tests. 

In the second stage of the study, experiments were carried out to identify the effect of the addition of acetic or lactic acid against *P. brenneri* MGM3 on Primo sale cheese: during the first trial (Figure 1), no significant differences were revealed in bacterial growth between control samples and treated ones, when applying surface dipping with acetic or lactic acid solutions with a concentration equal to the MIC determined in the first part of the study or doubling this concentration. No strain inactivation or growth reduction was observed, independently form the treatment (as shown by the values obtained at t_3_). A rapid growth at 6 °C was revealed in all the four series, with increases from t_0_ after 4 days of +3.28, +2.83, +3.08, +3.20 and +3.36 Log CFU/g in CTRL, Ac1, Ac2, Lac1 and Lac2, respectively. pH showed a slight decrease from t_0_ to the end of the trial (t_96_) from on average 6.31 to 6.02 (CTRL), 6.02 (Ac1), 6.08 (Ac2), 6.15 (Lac1) and 6.06 (Lac2), respectively. Such similarity among all the solutions tested, including those with a concentration corresponding to the double of the MIC obtained in broth, demonstrated the importance of the buffer activity of the food matrix. Moreover, the environmental conditions of a real cheese substrate can give some protection to the bacteria present on the surface, supplying numerous sites for attaching and growing; specific surface microenvironments can act as niches where the microorganisms can survive to harsh conditions (salt concentration, low temperature, low pH) and grow. Thus, these concentrations failed to prevent the growth of *P. brenneri*. in a real substrate condition.

In the second trial, the effect of the two organic acids against *P. brenneri* MGM3 growth on Primo sale cheese at 6 °C at higher concentrations were investigated (three times the MICs obtained in broth). The results obtained (Figure 2) showed a significantly lower growth in Ac3 (149.92 mM) samples if compared to control ones (*p* < 0.01): starting from an initial concentration of 3 Log CFU/g, a faster increase was observed in CTRL samples, reaching a final load of 7.39 Log CFU/g while a delayed and slower growth in Ac3 series led to a final load not overcoming 6 Log CFU/g. At the same conditions, Lac3 (133.20 mM) samples showed a significantly lower growth if compared to CTRL samples (*p* < 0.05), reaching a final value of 6.54 Log CFU/g. Despite the lower value obtained with the addition of acetic acid if compared to lactic acid, the comparison of the two treated series did not show any significant difference. In this trial, cheese pH were shown to be affected by the addition of organic acid solutions: a slight and gradual decrease from t0 (6.30) to t72 (6.18, CTRL), (6.01, Ac3), (5.12 Lac3), was reached.

A conditioning of the substrate by dipping the surface of cheese in a solution of organic acids after production could be a useful hurdle to reduce the increase of *P. brenneri*. The growth of this spoilage agent was not completely avoided thus no inactivation was observed, but it was slowed down significantly, especially by acetic acid, obtaining a difference of 1.57 Log CFU/g after three days of storage in Ac3 series. This outcome could be exploited in the dairy industry to delay the spoilage by *Pseudomonas*: it could be advantageous in terms of shelf-life extension, taking in mind the threshold associated with lower product acceptability (6 Log CFU/g) and the value that corresponds to negative sensorial properties of the product (7 Log CFU/g) [32,33]. A reduction in *Pseudomonas* growth could allow to reach later these two limits. 

When defining an application protocol, the producer should balance the needs to reach growth inhibition (and a consequently longer shelf life) with the minimization of the effect on the product taste, thus maintaining the original characteristics of this popular cheese. In this light, further research is needed, including the parallel evaluation of different combinations with concentration scales of lactic acid and acetic acid. This approach could allow to define the relative impact of each acid on both sensorial characters (evaluated by panellists) and distance from the microbial growth boundary.

## 4. Conclusions

The surface of fresh cheeses is characterized by permissive conditions for growth of spoilage microorganisms. Moreover, during production, contamination cannot be completely avoided; in this light, *Pseudomonas* spp. represents the main impediment to extend fresh cheeses’ shelf life. This study shows the potential inhibitory activity of acetic or lactic dipping as final surface treatment for Primo sale cheese; nevertheless, such treatment is intended to be applied as a part of an integrated hygiene plan, in order to couple a low initial microbial load with a scarce growth through the shelf life. 

## Figures and Tables

**Figure 1 foods-10-00891-f001:**
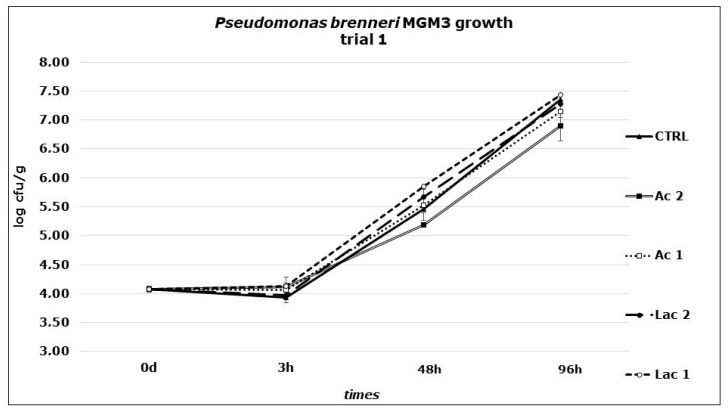
*P. brenneri* MGM3 counts obtained in Primo sale cheese during the first challenge test performed at 6 °C. Dipping: Ac1: acetic acid, 49.96 mM; Ac2: acetic acid, 99.92 mM; Lac1: lactic acid, 44.40 mM; Lac2: lactic acid, 88.80 mM; CTRL: distilled water.

**Figure 2 foods-10-00891-f002:**
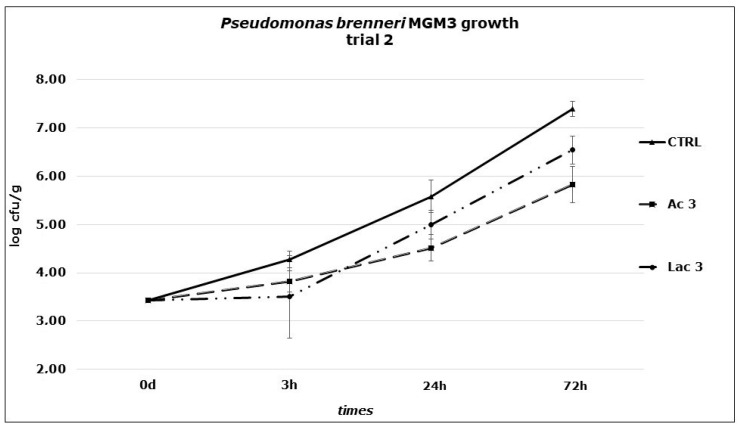
*P. brenneri* MGM3 counts obtained in Primo sale cheese during the second challenge test performed at 6 °C. Dipping: Ac3: acetic acid, 149.89 mM; Lac3: lactic acid, 133.20 mM; CTRL: distilled water.

**Table 1 foods-10-00891-t001:** Optical Density of *P. brenneri* MGM3 suspensions in Brain Heart Infusion (BHI) broth at 30 °C in presence of different concentrations of lactic and acetic acid (the value indicates the difference towards the respective blank sample). MIC detected are written in bold.

Acetic Acid	t_0_	t_24_	t_48_	t_72_
	Average ± Std.Dev	Average ± Std.Dev	Average ± Std.Dev	Average ± Std.Dev
49.96 mM	−0.02 ± 0.03	0.01 ± 0.04	0.01 ± 0.04	0.01 ± 0.04
24.98 mM	−0.05 ± 0.01	−0.03 ± 0.42	0.28 ± 0.42	0.57 ± 0.16
12.49 mM	−0.04 ± 0.01	0.12 ± 0.01	1.54 ± 0.01	1.76 ± 0.13
5.00 mM	−0.12 ± 0.01	0.20 ± 0.18	1.55 ± 0.18	1.55 ± 0.14
2.50 mM	−0.09 ± 0.01	0.28 ± 0.02	1.47 ± 0.02	1.61 ± 0.03
CTRL	0.01 ± 0.09	0.35 ± 0.01	1.65 ± 0.01	1.72 ± 0.06
**Lactic Acid**	**t_0_**	**t_24_**	**t_48_**	**t_72_**
	**Average ± std.dev**	**Average ± std.dev**	**Average ± std.dev**	**Average ± std.dev**
44.40 mM	−0.06 ± 0.03	−0.04 ± 0.02	−0.05 ± 0.02	0.01 ± 0.01
22.20 mM)	−0.08 ± 0.01	−0.03 ± 0.04	1.42 ± 0.01	1.79 ± 0.06
11.10 mM	−0.08 ± 0.05	0.24 ± 0.03	1.60 ± 0.04	1.65 ± 0.04
5.55 mM	−0.22 ± 0.26	0.35 ± 0.01	1.58 ± 0.15	1.56 ± 0.12
2.78 mM	0.24 ± 0.39	0.35 ± 0.07	1.57 ± 0.04	1.80 ± 0.09
CTRL	0.01 ± 0.09	0.35 ± 0.05	1.65 ± 0.01	1.72 ± 0.06

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
