# Peer review of "Pseudomonas spp.: Are Food Grade Organic Acids Efficient against These Spoilage Microorganisms in Fresh Cheeses?"

_foods, 2021, doi:10.3390/foods10040891_

Round 1

Reviewer 1 Report

Overview

The paper deals with the determination of Minimal Inhibiting Concentrations (MICs) of lactic and acetic acids towards Pseudomonas fluorescens and also the evaluation of the efficacy of a cheese surface treatment with these two organic acids. Minimal Inhibiting Concentrations (49.96 and 44.40 mM of acetic and lactic acid, respectively) were determined in BHI broth at 30 ºC. Semi-soft cow’s milk “Primo sale” cheeses were inoculated with P. fluorescens and then dipped into different acid solutions and stored at 6 ºC. A significantly lower growth was revealed at the highest concentrations tested for both acetic and lactic acids (149.89 and 133.20 mM, respectively) when compared to control samples. It is concluded that a conditioning of “Primo sale” surface with these organic acid solutions could be a useful hurdle for P. fluorescens inhibition and shelf life extension.

In my opinion, it may be questioned whether the research is broad enough to be able to be presented as an original article. The methodology is very simple, and no information is provided about the Pseudomonas fluorescens strain used in the study. The text could be shortened to a great extent, since the Results and discussion section includes some information that is not relevant to the study, such as that related to the first trial (lower concentracions of organic acids) performed on cheeses stored at 6 ºC. Furthermore, no mention is made of whether the treatments with the highest acetic and lactic acid concentrations could affect the sensory characteristics of the cheese, particularly if the cheese rind is an edible part.

General comments

- The English language should have been revised by a native English speaker with some knowledge of the subject.

- The Discussion and the Conclusion sections should be shortened.

- The number of tables and figures should be reduced.

Detailed comments

Abstract

- p. 1. Line 14. Avoid starting the sentence with an abbreviation.

- p. 1. Lines 15–16. Perhaps “inhibition test” would be more appropriate than “challenge test”.

- p. 1. Write “Primo sale” with the initial letter capitalized, here and throughout the text.

- p. 1. Line 22. Change “to contrast spoilage” to e.g. “to fight spoilage”.

Introduction

- p. 2. Line 49. Fresh cheeses made by mixed or predominantly acid coagulation (acid-curd cheeses) do not have a neutral pH.

- p. 2. Lines 60–62. Change “Aim” to e.g. “The first aim”. What specifically was the “fresh cheese matrix” from which P. fluorescens was isolated?

- p. 2. Lines 62–64. Rewrite the sentence avoiding the colon. Is “Primo sale” a cheese with a Protected Designation of Origin? Is it perhaps a fresh cheese made from cow's milk and salted? How is it salted?

Materials and methods

- p. 2. What is “Jenway 6105”? Is it perhaps a spectrophotometer? It should be specified.

- p. 2. Line 76. Avoid starting the sentence with a number written in figures. Change to e.g.: “Ten millilitre tubes…”

- p. 2. Line 90. What does “early stage of maturation” mean? Does that refer to “short-ripened” cheese? What is the average weight of Primo sale cheeses?

- p. 2. Line 94. Were the Primo sale cheeses sampled semi-soft or soft cheeses? What was their moisture in fat free basis content?

- p. 3. Lines 99–103. Delete the sentence “the culture was harvested… to a concentration around 106 CFU/mL” and change to e.g.: “as desceibed in the previous subsection”.

- p. 3. Line 112. CH?

- p. 3. Lines 116–117. What is the meaning of “pre-chilled” sterile saline? Provide the commercial data for “Stomacher 400”.

- p. 3. Line 119. Change “E” to “Spain”.

- p. 3. Line 121. How was the pH of cheese surface measured? Was the measurement made in an aqueous suspension? Change “I” to “Italy”.

- p. 3. Lines 123–127. If no favorable results were obtained in the “first challenge test”, the two trials should be described together (specifying all concentrations of the acidic solutions), and not differentiate.

- p. 3. Line 130. Provide the country for “SAS” software (USA?).

Results and discussion

- p. 3. Line 137. Change “thanks to” to e.g. “due to”.

- p. 3. Line 141. Change “Fresh cheese” to “Fresh cheeses”. Write “Mozzarella” with the initial letter capitalized.

- p. 3. Line 145. Change “Na2-EDTA” to “EDTA-Na2”.

- p. 4. Line 154. Write "pseudomonads" with the initial letter in lowercase.

- p. 4. Line 157. Change “solubilises” to “solubilisates”.

- p. 4. Lines 178–179. Change “Log” to “Log CFU/g”. The absence of significant differences between the control and treated samples has been mentioned previously (Lines 172–173).

- p. 4. Line 183. Change “importance the buffer activity” to “importance of the buffer activity”.

- p. 5. Line 186. Table 1. What is the meaning of “Bold: MIC detected” in the table legend? (MIC detected are written in bold numbers?).

- p. 5. Line 189. Figure 1. Change “AC2” to “Ac2”.

- p. 6. Lines 192–193. Figure 2. Check for: “For the legend, see Figure 1” (The meanings of the abbreviations are described in Figure 1).

- p. 6. Lines 194–207. The data obtained in the two trials should be showed and discussed together. Figures would not be necessary for the data from the first trial.

- p. 7. Lines 212–213. Figure 4. Check for: “For the legend, see Figure 3”.

- p. 7. Line 216. Change “replication” to e.g. “proliferation” or “increase”.

Conclusions

- p. 8. Lines 225–228. The sentence is not a conclusion of the study, but should appear in the Introduction section. Change “spoilage growth” to e.g. “growth of spoilage microorganisms”.

- p. 8. Lines 234–235. Which are the “original characteristics of this popular cheese”? They have not been mentioned in the text. What would be the estimated cost of applying high concentrations of lactic or acetic acids to achieve adequate growth inhibition of Pseudomonas spp.?

Author Response

Reviewer 1

Furthermore, no mention is made of whether the treatments with the highest acetic and lactic acid concentrations could affect the sensory characteristics of the cheese, particularly if the cheese rind is an edible part.

The aim of the study was to evaluate the potentiality of the adjunct of organic acid solutions to contrast Pseudomonas spp. growth, as the application of such treatment could be particularly easy for the producer This evaluation step is needed to drive further research facing the impact on the sensorial characteristics of the product, e.g. with the evaluation of organic acid salts, that can have a lower impact but also a potentially lower activity.

General comments

- The English language should have been revised by a native English speaker with some knowledge of the subject. This was done

- The Discussion and the Conclusion sections should be shortened. We did it

- The number of tables and figures should be reduced. We did it

Detailed comments

Abstract

- p. 1. Line 14. Avoid starting the sentence with an abbreviation. This was done

- p. 1. Lines 15–16. Perhaps “inhibition test” would be more appropriate than “challenge test”. This was done

- p. 1. Write “Primo sale” with the initial letter capitalized, here and throughout the text. This was done

- p. 1. Line 22. Change “to contrast spoilage” to e.g. “to fight spoilage”. This was done

Introduction

- p. 2. Line 49. Fresh cheeses made by mixed or predominantly acid coagulation (acid-curd cheeses) do not have a neutral pH. We do not specify in this part cheeses made by mixed or predominantly acid coagulation, we only specific fresh cheese

- p. 2. Lines 60–62. Change “Aim” to e.g. “The first aim”. What specifically was the “fresh cheese matrix” from which P. fluorescens was isolated? This was done, we also specified the matrix

- p. 2. Lines 62–64. Rewrite the sentence avoiding the colon. Is “Primo sale” a cheese with a Protected Designation of Origin? Is it perhaps a fresh cheese made from cow's milk and salted? How is it salted? We rephrased the sentence

Materials and methods

- p. 2. What is “Jenway 6105”? Is it perhaps a spectrophotometer? It should be specified. This was done

- p. 2. Line 76. Avoid starting the sentence with a number written in figures. Change to e.g.: “Ten milliliter tubes…” This was done

- p. 2. Line 90. What does “early stage of maturation” mean? Does that refer to “short-ripened” cheese? What is the average weight of Primo sale cheeses? Details were included

- p. 2. Line 94. Were the Primo sale cheeses sampled semi-soft or soft cheeses? What was their moisture in fat free basis content? We corrected this information, as primo sale has moisture >80% we defined as fresh cheese

- p. 3. Lines 99–103. Delete the sentence “the culture was harvested… to a concentration around 106 CFU/mL” and change to e.g.: “as described in the previous subsection”. This was done

- p. 3. Line 112. CH? CH= Switzerland

- p. 3. Lines 116–117. What is the meaning of “pre-chilled” sterile saline? Maintained at refrigeration temperature before use Provide the commercial data for “Stomacher 400”. These details were included

- p. 3. Line 119. Change “E” to “Spain”. This was done

- p. 3. Line 121. How was the pH of cheese surface measured? Was the measurement made in an aqueous suspension? Change “I” to “Italy”. This was done, details were included

- p. 3. Lines 123–127. If no favorable results were obtained in the “first challenge test”, the two trials should be described together (specifying all concentrations of the acidic solutions), and not differentiate. We do not agree as the starting concentration and pH are different, thus should be considered as two independent studies

- p. 3. Line 130. Provide the country for “SAS” software (USA?). This was done

Results and discussion

- p. 3. Line 137. Change “thanks to” to e.g. “due to”. This was done

- p. 3. Line 141. Change “Fresh cheese” to “Fresh cheeses”. Write “Mozzarella” with the initial letter capitalized. This was done

- p. 3. Line 145. Change “Na2-EDTA” to “EDTA-Na2”. This was done

- p. 4. Line 154. Write "pseudomonads" with the initial letter in lowercase. This was done

- p. 4. Line 157. Change “solubilises” to “solubilisates”. This was done

- p. 4. Lines 178–179. Change “Log” to “Log CFU/g”. The absence of significant differences between the control and treated samples has been mentioned previously (Lines 172–173). This was done

- p. 4. Line 183. Change “importance the buffer activity” to “importance of the buffer activity”. This was done

- p. 5. Line 186. Table 1. What is the meaning of “Bold: MIC detected” in the table legend? (MIC detected are written in bold numbers?). yes

- p. 5. Line 189. Figure 1. Change “AC2” to “Ac2”. This was done

- p. 6. Lines 192–193. Figure 2. Check for: “For the legend, see Figure 1” (The meanings of the abbreviations are described in Figure 1). This was done

- p. 6. Lines 194–207. The data obtained in the two trials should be showed and discussed together. Figures would not be necessary for the data from the first trial. The two trials are independent, cheese has although small, a difference in starting inoculation concentration and pH; we reduced the number of the figures, according to reviewer 2

- p. 7. Lines 212–213. Figure 4. Check for: “For the legend, see Figure 3”. This was done

- p. 7. Line 216. Change “replication” to e.g. “proliferation” or “increase”. This was done

Conclusions

- p. 8. Lines 225–228. The sentence is not a conclusion of the study, but should appear in the Introduction section. Change “spoilage growth” to e.g. “growth of spoilage microorganisms”. This was done

- p. 8. Lines 234–235. Which are the “original characteristics of this popular cheese”? They have not been mentioned in the text. The meaning ids that the inclusion of a dipping in an acid solution should not change the original sensorial properties of the cheese.

What would be the estimated cost of applying high concentrations of lactic or acetic acids to achieve adequate growth inhibition of Pseudomonas spp.? it is negligible (we estimate less than 10 cent of euro for kg of primo sale cheese)

Reviewer 2 Report

The manuscript show interesting study, well written but unfortunately poor presentatation of data and discussion that need to be improved.

L37; gram shouuld be Gram

L39 move (and) after the parentheses

L166 delete :different statements were found in the cited studies. Rephrase the sentence and refer to the references

The authors should discuss in depth the reason for no strain inactivation was observed in the second stage of the study. Indicate the possible reasons and differences between stages

P5 Table 1: show the values as average ±SD; please don’t separate in two columns

Fig 1 and 2 is are recommended to be combined in one figure as 1a , b

Fig 3 and 4 is are recommended to be combined in one figure as 2a , b

The presentation and Quality of figures need to be improved. Provide line at X and Y axes, unify the range among figures.  And confirm the range of error bars. See for example data at fig 4 after 3h

Authors should clarify the effect of acetic acid and Lactic acid for human consumption, allowed range in the MS

The authors should clarify and compare other characteristics of cheese after the challenge test especially organoleptic, appearance, texture, ….etc

The Ms is missing for more in depth discussion.

Author Response

Reviewer 2

L37; gram should be Gram This was done

L39 move (and) after the parentheses This was done

L166 delete: different statements were found in the cited studies. Rephrase the sentence and refer to the references This was done

The authors should discuss in depth the reason for no strain inactivation was observed in the second stage of the study. Indicate the possible reasons and differences between stages a growth inhibition although not an inactivation was observed at the highest concentration of acetic acid; this was included

P5 Table 1: show the values as average ±SD; please don’t separate in two columns This was done

Fig 1 and 2 is are recommended to be combined in one figure as 1a , b  Fig 3 and 4 is are recommended to be combined in one figure as 2a , b We deleted figure 2 and 4 as suggested by reviewer 1

The presentation and Quality of figures need to be improved. Provide line at X and Y axes, unify the range among figures.  And confirm the range of error bars. See for example data at fig 4 after 3h.  We improved the figures.

Authors should clarify the effect of acetic acid and Lactic acid for human consumption, allowed range in the MS We clarified

The authors should clarify and compare other characteristics of cheese after the challenge test especially organoleptic, appearance, texture,….etc   The aim of the study was to evaluate the potentiality of the adjunct of organic acid solutions to contrast Pseudomonas spp. growth, as the application of such treatment could be particularly easy for the producer This evaluation step is needed to drive further research facing the impact on the sensorial characteristics of the product.

The Ms is missing for more in depth discussion. We tried to improve

Reviewer 3 Report

Is the Pseudomonas strain you are investigating deposited in the collection?

"A strain of P. fluorescens, previously isolated from a fresh dairy product, was used for this trial" - is there any article of yours describing this strain? If yes, you should add the citation in this place. If not, then it should be written how it was isolated and how it was identified.

Acetic acid has a rather specific smell and taste, wouldn't its addition to dairy products negatively affect their session properties?

"When defining an application protocol, a combination of the organic acids tested should be further studied: this could allow the producer to balance the needs to reach growth inhibition (and a consequently longer shelf life) and to minimize the effect on the product taste, thus maintaining the original characteristics of this popular cheese." - what exactly do you mean? Maybe you should use some statistical method, e.g. DoE, to choose the right combinaton of acids and their concentration so as to minimize their negative impact on the taste or texture of cheeses.

Minor spell check required

Author Response

Reviewer 3

Is the Pseudomonas strain you are investigating deposited in the collection? The strain is part of a private collection of the lab

"A strain of P. fluorescens, previously isolated from a fresh dairy product, was used for this trial" - is there any article of yours describing this strain? If yes, you should add the citation in this place. If not, then it should be written how it was isolated and how it was identified. This was included

Acetic acid has a rather specific smell and taste, wouldn't its addition to dairy products negatively affect their session properties?

The aim of the study was to evaluate the potentiality of the adjunct of organic acid solutions to contrast Pseudomonas spp. growth, as the application of such treatment could be particularly easy for the producer This evaluation step is needed to drive further research facing the impact on the sensorial characteristics of the product, e.g. with the evaluation of organic acid salts, that can have a lower impact but also a potentially lower activity.

"When defining an application protocol, a combination of the organic acids tested should be further studied: this could allow the producer to balance the needs to reach growth inhibition (and a consequently longer shelf life) and to minimize the effect on the product taste, thus maintaining the original characteristics of this popular cheese." - what exactly do you mean? Maybe you should use some statistical method, e.g. DoE, to choose the right combination of acids and their concentration so as to minimize their negative impact on the taste or texture of cheeses. We tried to clarify the concept in the last part of the discussion

Round 2

Reviewer 1 Report

General comments

I think that the manuscript has been significantly improved, including English writing. However, I still believe that the paper lacks sufficient entity to be published as an original article, although it could be published as a research note or a short communication. Since this last possibility is not taken into account by the journal, I think that the Editor should finally decide if the manuscript can be accepted or not.

Some final corrections must be made imperatively. I provide the authors with the new comments and corrections. I also hope they will be very watchful of correcting any other deficiencies or errors they may encounter if the Editor decides to accept the article for publication.

Detailed comments

- Lines 48–49. As I stated in my first review, there are fresh cheeses made by a predominantly lactic or acid coagulation like Quark or Cottage that do not have a neutral pH. You should specify e.g. “fresh cheeses made by enzymatic coagulation” or “rennet-curd fresh cheeses”.

- Line 68. Write “Mozzarella” with the initial capitalized.

- Lines 73–74. Change to e.g. “… at 540 nm determined with a spectrophotometer (Jenway 6105, …).”

- Line 115. Change “Stomacher BagMixer® 400W” to “peristaltic blender BagMixer® 400W”. “Stomacher” is the name of a trademark.

- Lines 120–121. Change to e.g. “pH meter equipped with a penetration electrode (Amel, Milan, Italy).”, or perhaps “pH meter equipped with a microelectrode (Amel, Milan, Italy).” However, the correct way to measure the pH of the cheese surface should have been to prepare a surface homogenate in distilled water. The measurements should have been carried out in triplicate at least.

- Line 189. Change “acidification” to e.g. “low pH”.

- Figures 1 and 2. Times should be shown in hours (0 h, 3 h, 24 h, 72 h). Change “log cfu/g” to “Log CFU/g” as in the text. Be consistent.

- Line 208. Change to e.g. “and a gradual decrease was observed from t0 (6.30) to t72, when values of 6.18 (CTRL), … were reached.”

- Line 236. Insert “acid” after “lactic”.

Author Response

Lines 48–49. As I stated in my first review, there are fresh cheeses made by a predominantly lactic or acid coagulation like Quark or Cottage that do not have a neutral pH. You should specify e.g. “fresh cheeses made by enzymatic coagulation” or “rennet-curd fresh cheeses”. This was done

- Line 68. Write “Mozzarella” with the initial capitalized. This was done

- Lines 73–74. Change to e.g. “… at 540 nm determined with a spectrophotometer (Jenway 6105, …).” This was done

- Line 115. Change “Stomacher BagMixer® 400W” to “peristaltic blender BagMixer® 400W”. “Stomacher” is the name of a trademark. This was done

- Lines 120–121. Change to e.g. “pH meter equipped with a penetration electrode (Amel, Milan, Italy).”, or perhaps “pH meter equipped with a microelectrode (Amel, Milan, Italy).” However, the correct way to measure the pH of the cheese surface should have been to prepare a surface homogenate in distilled water. The measurements should have been carried out in triplicate at least. This was done

- Line 189. Change “acidification” to e.g. “low pH”. This was done

- Figures 1 and 2. Times should be shown in hours (0 h, 3 h, 24 h, 72 h). Change “log cfu/g” to “Log CFU/g” as in the text. Be consistent. This was done

- Line 208. Change to e.g. “and a gradual decrease was observed from t0 (6.30) to t72, when values of 6.18 (CTRL), … were reached.” This was done

- Line 236. Insert “acid” after “lactic”. This was done

Reviewer 2 Report

The authors have addressed almost raised comments.

Author Response

thank you

Reviewer 3 Report

The authors introduced the proposed changes and responded to comments. Everything is ok.

Author Response

thank you
